# Newly Proposed Dose of Daclatasvir to Prevent Lethal SARS-CoV-2 Infection in Human Transgenic ACE-2 Mice

**DOI:** 10.3390/v16121856

**Published:** 2024-11-29

**Authors:** Mayara Mattos, Carolina Q. Sacramento, André C. Ferreira, Natalia Fintelman-Rodrigues, Filipe S. Pereira-Dutra, Caroline Souza de Freitas, João S. M. Gesto, Jairo R. Temerozo, Aline de Paula Dias Da Silva, Mariana T. G. Moreira, Rafael S. C. Silva, Gabriel P. E. Silveira, Douglas P. Pinto, Heliana M. Pereira, Laís B. Fonseca, Marcelo Alves Ferreira, Camilla Blanco, João P. B. Viola, Dumith Chequer Bou-Habib, Patrícia T. Bozza, Thiago Moreno L. Souza

**Affiliations:** 1Laboratório de Imunofarmacologia, Instituto Oswaldo Cruz (IOC), Fundação Oswaldo Cruz (Fiocruz), Rio de Janeiro 21040-361, RJ, Brazil; maymattos03@gmail.com (M.M.); carol.qsacramento@gmail.com (C.Q.S.); andre.bio2009@gmail.com (A.C.F.); nataliafintelman@gmail.com (N.F.-R.); filipe.spd@gmail.com (F.S.P.-D.); caroline23.freitas@gmail.com (C.S.d.F.); aline.paula@hotmail.com.br (A.d.P.D.D.S.); camillablanco.a@gmail.com (C.B.); pbozza@gmail.com (P.T.B.); 2National Institute for Science and Technology on Innovation in Diseases of Neglected Populations (INCT/IDPN), Center for Technological Development in Health (CDTS), Fundação Oswaldo Cruz (Fiocruz), Rio de Janeiro 21040-361, RJ, Brazil; malvesf68@gmail.com; 3Laboratório de Pesquisas Pré-Clínicas, Departamento de Ciências Biológicas, Universidade Iguaçu, Nova Iguaçu 26275-580, RJ, Brazil; 4SESI Innovation Center for Occupational Health, Rio de Janeiro 22735-280, RJ, Brazil; jgesto@firjan.com.br (J.S.M.G.); dumith.chequer@gmail.com (D.C.B.-H.); 5Laboratório de Pesquisas Sobre o Timo, Instituto Oswaldo Cruz (IOC), Fundação Oswaldo Cruz (Fiocruz), Rio de Janeiro 21040-360, RJ, Brazil; jairo.jrt@gmail.com; 6National Institute for Science and Technology on Neuroimmunomodulation (INCT/NIM), Instituto Oswaldo Cruz (IOC), Fundação Oswaldo Cruz (Fiocruz), Rio de Janeiro 21040-360, RJ, Brazil; 7Equivalence and Pharmacokinetics Service (SEFAR), Vice-Presidency of Production and Innovation in Health (VPPIS), Fundação Oswaldo Cruz (Fiocruz), Rio de Janeiro 21040-360, RJ, Brazil; mariana.moreira@fiocruz.br (M.T.G.M.); rafael.caetano@fiocruz.br (R.S.C.S.); gabriel.silveira@fiocruz.br (G.P.E.S.); douglas.pinto@fiocruz.br (D.P.P.); heliana.pereira@fiocruz.br (H.M.P.); lais.fonseca@fiocruz.br (L.B.F.); 8Program of Immunology and Tumor Biology, Brazilian National Cancer Institute (INCA), Rio de Janeiro 20230-130, RJ, Brazil; jpbviola@gmail.com

**Keywords:** SARS-CoV-2, anti-HCV drugs, exonuclease, error prone, K18-hACE2 mice

## Abstract

Coronavirus disease 2019 (COVID-19) still causes death in elderly and immunocompromised individuals, for whom the sustainability of the vaccine response may be limited. Antiviral treatments, such as remdesivir or molnupiravir, have demonstrated limited clinical efficacy. Nirmatrelvir, an acute respiratory syndrome coronavirus 2 (SARS-CoV-2) major protease inhibitor, is clinically effective but has been associated with viral rebound and antiviral resistance. It is thus necessary to study novel and repurposed antivirals for the treatment of COVID-19. We previously demonstrated that daclatasvir (DCV), an inhibitor of the hepatitis C virus (HCV) NS5A protein, impairs SARS-CoV-2 replication by targeting viral RNA polymerase and exonuclease, but the doses of DCV used to inhibit the new coronavirus are greater than the standard human plasma exposure for hepatitis C. Because any potential use of DCV against SARS-CoV-2 would be shorter than that reported here and short-term toxicological studies on DCV show that higher doses are tolerable, we searched for doses of DCV that could protect transgenic mice expressing the human ACE2 receptor (K18-hACE-2) from lethal challenge with SARS-CoV-2. We found that a dose of 60 mg/kg/day provides this protection by reducing virus replication and virus-induced lung insult. This dose is tolerable in different animal models. Taken together, our data provide preclinical evidence that can support phase I clinical trials to confirm the safety, tolerability, and pharmacokinetics of new doses of daclatasvir for a short duration in humans to further advance this compound’s utility against COVID-19.

## 1. Introduction

Coronavirus disease 2019 (COVID-19), caused by severe acute respiratory syndrome coronavirus 2 (SARS-CoV-2), has resulted in more than 6.9 million confirmed deaths globally and has left a legacy of long-term morbidities that require ongoing attention from healthcare systems [1]. In the post-pandemic era, COVID-19-associated critical illness and deaths occur mainly in elderly individuals and individuals with comorbidities worldwide [2] because of the limited sustainability of the immune response to vaccines [3] and viral rebound associated with the use of the antiviral Paxlovid [4]. In addition, the emergence of antiviral-resistant mutants, such as those resistant to remdesivir (RDV) [5] and nirmatrelvir [6,7], has motivated the development or repurposing of antivirals.

Like other positive-sense RNA viruses, SARS-CoV-2 possesses an RNA-dependent RNA polymerase (RdRP), represented by nonstructural protein (nsp) 12, which is assisted by the viral cofactors nsp7 and nsp8 [8]. This enzyme associates with other viral nonstructural proteins to replicate and transcribe the virus genome. Among these viral proteins, nsp14/10 allows the SARS-CoV-2 genome to be proofread during replication because of its exonuclease (ExoN) function. ExoN will rectify incorrect base pairing, which occurs during RNA synthesis by the viral RNA polymerase, and atypical nucleotides, such as antiviral nucleotide analogs taken up by the viral polymerase [9,10]. Thus, antivirals that target RdRp and ExoN could potentially be useful against SARS-CoV-2.

We previously demonstrated that daclatasvir, a clinically approved inhibitor of hepatitis C virus (HCV) nonstructural protein 5a (NS5A), inhibits SASR-CoV-2 RdRp and ExoN [11,12]. However, daclatasvir’s in vitro pharmacological ability to inhibit SARS-CoV-2 was greater than that of human plasma exposure [12,13]. At the standard human anti-HCV dose of 60 mg/day, daclatasvir inhibits 50 to 90% of SARS-CoV-2 replication [12]. To achieve more than 90% inhibition, higher doses are necessary [12]. For the preclinical development of daclatasvir against HCV, such as in nonhuman primates, animals were treated daily for 4–9 months, and the no-observable-effect level (NOEL) was determined to be 15 mg/kg, followed by the clinically approved human dose of 60 mg/day (or 1 mg/kg/day on the basis of the human reference body weight for pharmacokinetics studies) [13,14]. We rationalized that repurposing daclatasvir against COVID-19 would not require a treatment regimen as long as that for HCV because the natural history of SARS-CoV-2 infection is shorter. Thus, to further advance the preclinical repurposing of daclatasvir against COVID-19 beyond the in vitro and molecular studies performed by us [11,12], we evaluated whether mice lethally challenged with SARS-CoV-2 could survive upon receiving new doses of daclatasvir.

## 2. Materials and Methods

### 2.1. Reagents

The antiviral daclatasvir (drug and analytical standard) used in the validation and conductance of the study was received as a donation from Microbiologica Química-Farmacêutica LTDA (Rio de Janeiro, Brazil). The internal standard carbamazepine (IS) was obtained from United States Pharmacopeia (USP). The inhibitor was dissolved in 100% dimethylsulfoxide (DMSO Hymenax; Sigma–Aldrich/Merck, St. Louis, MO, USA). The materials for cell culture were purchased from Thermo Scientific Life Sciences (Grand Island, NY, USA), unless otherwise mentioned. ELISA kits for the cytokines TNF-α, IL-6, and KC were purchased from R&D Bioscience, and the materials used for genome sequencing were purchased from MGI, Shenzhen, China.

### 2.2. Cells and Virus

African green monkey kidney cells (Vero, subtype E6, ATCC, Manassas, Virginia) were cultured in high-glucose DMEM supplemented with 10% fetal bovine serum (FBS; HyClone, Logan, Utah), 100 U/mL of penicillin and 100 μg/mL of streptomycin (P/S) at 37 °C in a humidified atmosphere with 5% CO2. The SARS-CoV-2 gamma variant, also referred to as the P1 lineage (EPI_ISL_1060902, hCoV-19/Brazil/AM-L70-71-CD1739/2020), was cultured from nasopharyngeal swabs of confirmed COVID-19 cases via Vero E6 cells, which were used for SARS-CoV-2 propagation and titration.

### 2.3. Animals

The animals used in these experiments, both Swiss Webster and transgenic male mice expressing the human ACE-2 receptor (K18-hACE2), were obtained from the Oswaldo Cruz Foundation breeding colony. The animals were maintained with free access to food and water at 29–30 °C under a controlled 12 h light/dark cycle. The experiments were performed during the light phase of each cycle. All in vivo experiments with SARS-CoV-2 were performed in an animal biosafety level 3 (ABSL-3) multiuser facility at the Brazilian National Cancer Institute (INCa), following WHO guidelines. The animal welfare guidelines of the Ethics Committee of Animal Experimentation (CEUA) licensed the use of Swiss Webster and K18-hACE2 mice under the codes #CEUA-L006/20 (approved on 20 July 2020) and #CEUA-INCa-L005/2021 (approved on 3 May 2021), respectively.

### 2.4. Experimental Infection In Vivo

K18-hACE2 mice (10–12 weeks old) were anesthetized with 60 mg/kg of ketamine and 4 mg/kg of xylazine and inoculated intranasally with 10^5^ PFU of the SARS-CoV-2 P.1 lineage. This lineage was able to achieve 100% lethality in a short time frame [15]. Eight mice were used per experimental group: mock (noninfected), SARS-CoV-2-infected without treatment (nil), and SARS-CoV-2-infected and treated with daclatasvir at 10, 30, or 60 mg/kg. For proof of principle, we used not only the target dose of 60 mg/kg but also suboptimal doses. The treatments were started overnight (12 h) after infection by oral gavage and continued for the next 5 days. The animals were monitored daily for survival, body weight, and clinical signs of infection, such as weight loss, reduced behavioral activity and exploration, eye closure or tearing, piloerection, posture, and respiration [16]. Euthanasia was performed to alleviate animal suffering in cases of weight loss > 25%. For the collection of biological material, on the 6th day postinfection, euthanasia was performed subcutaneously (dorsal–anterior region of the animal) with a dose of 150 mg/kg of ketamine and 10 mg/kg of xylazine in a volume of 70 µL (using an insulin syringe with an 8 mm needle). Once the animal was fully anesthetized (pain sensitivity test performed by lightly pressing the paw), blood was collected via cardiac puncture (using a 3 mL syringe and a 22 G needle). After the completion of blood collection by cardiac puncture, the animals received a lethal dose of anesthetics (300 mg/kg of ketamine and 30 mg/kg of xylazine), and cardiac–respiratory arrest was confirmed via a stethoscope.

Bronchoalveolar lavage (BAL) from both lungs was performed by washing the lungs with 1 mL of cold PBS. After the centrifugation of the BAL fluid (500× *g* for 5 min), the pellet was used for total and differential leukocyte counts (diluted in Turk’s 2% acetic acid), and the supernatant was used for differential cell counts by cytospin and cytokine quantification through ELISA. Cytospin was performed via centrifugation at 350× *g* for 5 min and May–Grünwald–Giemsa staining. The lungs of the animals were collected after perfusion with 20 mL of saline solution. The lungs were sheared via a Potter homogenizer in the presence of 500 µL of a phosphatase and protease inhibitor cocktail (EDTA-free Roche Applied Science, Mannheim, Germany) and further homogenized for 30 s via an Ultra-Turrax Disperser T-10 basic IKA (Guangzhou, China). Lungs were assessed for viral load via quantitative RT–PCR and virus titration. Vero cells (2.0 × 10^4^ cells/well) in 96-well plates (Nalge Nunc Int., Rochester, NY, USA) were infected with serial log-based dilutions of supernatants from the lungs for 1 h at 37 °C with 5% CO_2_. Following incubation, medium containing 1.8% CMC and 5% FBS was added, and the cells were incubated at 37 °C with 5% CO_2_ for 72 h. The cells were then fixed with 10% formaldehyde in PBS and stained with a 0.04% crystal violet solution in 70% methanol. Virus titers were determined by counting the plaque-forming units (PFUs)/mL. In addition, histological analysis, unbiased sequencing and metatranscriptomic approaches were used.

### 2.5. Pharmacokinetics

Five-week-old Swiss Webster mice weighing approximately 30 g were maintained with free access to food and water when treated with daclatasvir at a dose of 60 mg/kg via oral (p.o.) gavage. Daclatasvir was initially dissolved in DMSO at a stock concentration of 100 mg/mL. After that, the stock was diluted to a working solution in PBS so that the final DMSO concentration was 1.25%, which was not harmful under our experimental conditions and according to the literature [17]. After oral administration, the animals were killed at different time points after treatment: 5, 10, 20, and 40 min and 1, 2, 3, 4, 6, 8, 10, 12, 14, and 16 h. At each time point, the plasma and lungs were immediately removed and processed for later analysis. Plasma was obtained by blood centrifugation at 8000× *g* for 15 min. A time of zero was obtained by analyzing the matrix pool (blood plasma) of untreated animals.

To determine the pharmacokinetic parameters, an average pharmacokinetic profile was generated on the basis of the mean plasma concentrations of the animals included at each time point interval. The pharmacokinetic parameters were determined via a noncompartmental model. The peak plasma concentration (C_max_) and time to peak concentration (T_max_) were obtained directly from the graphic of the plasma concentration versus time. The area under the plasma concentration versus time curve of time zero to the time of the last measurable concentration (AUC_last_) was obtained via the trapezoidal method, and the area under the total of time zero to infinity (AUC_∞_) was calculated via the equation AUC_last_ + (C_t_/λ), where C_t_ is the last concentration observed above the limit of quantification of the analytical method and “λ” is the apparent elimination constant obtained via the linear regression of the terminal phase points of the concentration versus the time curve after the logarithmic transformation of the plasma concentration data. The plasma elimination half-life (T_1/2_) was calculated as ln(2)/λ. The pharmacokinetic parameters were determined via Certara’s Phoenix WinNonlin^®^ 8.4 software.

### 2.6. Bioanalytical Method for the Analysis of Daclatasvir in Mouse Plasma

A sensitive method was developed and validated for the quantification of daclatasvir in mouse plasma containing the anticoagulant EDTA. The concentration range established was 50–15,000 ng/mL. Daclatasvir and internal standards were extracted from mouse plasma via the liquid–liquid extraction technique with tert-butyl methyl-ether (TBME). The samples were analyzed via liquid chromatography with a mobile phase composed of acetonitrile/ammonium formate (5 mmol/L) containing 0.1% formic acid (50/50—*v*/*v*) via an ACE C_8_ column (150 mm × 4.6 mm) and detected via mass spectrometry (SCIEX API 4000) with electrospray ionization (ESI+) in multiple reaction monitoring (MRM) mode. The monitored mass transitions were 739.4 > 565.3 for daclatasvir and 237.1 > 193.9 for the IS. This method was validated in accordance with the Brazilian regulatory agency (ANVISA) Bioanalytical Guidance of May 2012, which is also in accordance with European regulatory guidelines [18].

### 2.7. Quantification of Viral RNA

The viral RNA from samples collected from the mice was quantified via reverse transcription polymerase chain reaction (RT–PCR). Total RNA was extracted with the QIAamp Viral RNA Kit (Qiagen, Germantown, MD, USA) following the manufacturer’s instructions. Quantitative RT–PCR was conducted via the QuantiTect Probe RT–PCR Kit (Qiagen) on a StepOne Plus™ Real–Time PCR System (Thermo Fisher Scientific, Waltham, MA, USA). Amplifications were performed in 15 µL reaction mixtures containing 2× reaction mix buffer, 50 µM of each primer, 10 µM of the probe, and 5 µL of RNA template. The primers, probes, and cycling conditions recommended by the Centers for Disease Control and Prevention (CDC) protocol were used to detect SARS-CoV-2 (CDC 2020). The amplification of the housekeeping gene glyceraldehyde-3-phosphate dehydrogenase (GAPDH) was used as a reference for the number of cells. The cycle threshold (CT) values for this target were compared with those obtained with varying cell quantities (10^7^–10^2^) for calibration.

### 2.8. Unbiased SARS-CoV-2 Sequencing

The extracted and quantified SARS-CoV-2 RNA was subjected to unbiased sequencing via an MGI-g400 sequencer (MGI, Shenzhen, China) and a metatranscriptomic approach, as previously described [12]. In brief, total purified RNA from the samples was used for the construction of the libraries via the MGIEasy RNA Library Prep Set. The libraries were constructed through RNA fragmentation (250 bp), reverse-transcription and second-strand synthesis. The libraries were purified via MGIEasy DNA Clean Beads and then subjected to end repair, adaptor ligation, and PCR amplification steps. The samples were purified and quantified with a Qubit dsDNA HS Assay Kit using a Qubit 4.0 fluorometer (Invitrogen Waltham, MA, USA). The PCR products were homogeneously pooled and subjected to denaturation and circularization steps for transformation into a single-stranded circular DNA library. The purified libraries were quantified with a Qubit ssDNA Assay Kit using a Qubit 4.0 fluorometer. DNA nanoballs were generated by the rolling circle amplification of a pool and quantified and loaded onto the flow cell to be sequenced via the PE150 program (150-bp paired-end reads). The sequencing data were analyzed via the usegalaxy.org platform and then aligned via ClustalW via Mega 7.0 software.

### 2.9. SARS-CoV-2 Genome Assembly

The raw genomic data, FASTQ files, are available in the sequencing read archive (SRA) via bioproject PRJNA1161613. These sequences were demultiplexed and submitted to a customized Galaxy workflow for the analysis of paired-end amplicon data, along with the SARS-CoV-2 reference sequence (Wuhan-hu-1 isolate, GenBank MN908947.3). FASTQ sequences were preprocessed via FASTP v.0.20.1 to remove adapters and reads shorter than 50 bp (-l 50). Mapping and genome assembly were performed with BWA-MEM v. 0.7.17 with the default parameters. The output BAM files were filtered by quality (-q 20) and reformatted with SamTools view v.1.13 to exclude (-F) unmapped reads (and their mate pairs) and those not consisting of primary alignments. Additionally, reads were realigned to the reference genome with LoFreq v.2.1.5, adding indel qualities on the basis of the Dindel algorithm. Variants were called with iVar v.1.3.1 under an enhanced quality score (-q 30), allowing populations above 1% (-t 0.01) to be considered for the output VCF files. Consensus was called from the VCF file via BCFtools v. 1.10, allowing ambiguous bases.

### 2.10. Measurements of Inflammatory Mediators and Cell Death

The levels of IL-6, TNF-α, KC (keratinocyte-derived cytokine), and PF4 (platelet factor 4) in BAL samples from uninfected (MOCK), infected/untreated (NIL), and infected/treated animals were quantified via ELISA with specific kits following the manufacturer’s instructions (R&D Systems). Cell death was assessed by measuring LDH activity in the BALF.

### 2.11. Histological and Immunohistochemistry Procedures

Histological features associated with SARS-CoV-2-induced lung injury were analyzed in K18-hACE2 mice. Lung tissues were collected, fixed in 4% formaldehyde, dehydrated, and embedded in paraffin. Thin sections (5 μm) were obtained via a microtome and then fixed and stained with hematoxylin and eosin (H&E) for microscopic analysis. Morphological alterations in the tissue were observed and documented. The morphological alterations observed in the lung tissue were assessed using a previously published inflammatory scoring system (PMID: 34495692) [19]. For immunohistochemistry, the lung sections were deparaffinized, rehydrated, and subjected to antigen retrieval. To prevent nonspecific staining, the slides were incubated with normal guinea pig serum (Sigma–Aldrich, Burlington, MA, USA, cat #566400) for 30 min at room temperature. The sections were subsequently incubated with a primary antibody against dsRNA IgG2a mouse monoclonal antibody (Jena Biosciences, Jena, Germany, J2, cat # RNT-SCI-10010200, 1:500) at room temperature for 1 h and 30 min. After being rinsed with phosphate-buffered saline (PBS), the slides were incubated with labeled polymer-HRP (Peroxidase AffiniPure Goat Anti-Mouse IgG (H + L), Jackson ImmunoResearch, Inc., West Grove, PA, USA, cat # AB_10015289, 1:2000) for 30 min, according to the manufacturer’s instructions. The color reaction was developed via the use of 3,3′-diaminobenzidine tetrachloride (DAB) chromogen solution, and all the slides were counterstained with hematoxylin.

### 2.12. Statistical Analysis

The assays were performed in a blinded manner by one professional, codified, and then read by another professional. All experiments were carried out at least three independent times, including a minimum of two technical replicates in each assay. GraphPad Prism software 9.0 was used for scoring *p* values < 0.05 according to ANOVA. The statistical analyses specific to each software program used in the bioinformatics analysis are described above.

## 3. Results

### 3.1. Daclatasvir Dose to Protect hACE-2 Mice Against Lethal SARS-CoV-2 Infection

Our previous results demonstrated that daclatasvir inhibits both RNA polymerase and exonuclease activities [11,12], but higher doses would be necessary to reach plasma exposure to suppress virus replication [12]. On the basis of daclatasvir monography, a mouse equivalent dose of 60 mg/kg has a preclinical safety profile [14]. We thus infected hACE2 with SARS-CoV-2 intranasally and treated the animals daily via oral gavage with daclatasvir at 10, 30, and 60 mg/kg. The higher dose increased mouse survival, whereas suboptimal doses did not significantly affect this parameter (Figure 1A). The treatments also reduced the loss of body weight (Figure 1B) and the virus-induced score of clinical severity (Figure 1C).

At 60 mg/kg/day, daclatasvir reduced SARS-CoV-2 RNA levels and titers in the lungs of infected mice by up to 3 log10 (Figure 2A,B). Consistent with the described mechanism of ExoN inhibition, daclatasvir affected SARS-CoV-2 genomic stability (Figure 2C), leading to error-prone virus replication with greater G:A, T:C, and T:G ratios than those of viruses from untreated lungs.

### 3.2. Daclatasvir Reduces Virus-Induced Lung Damage

Severe SARS-CoV-2 infection leads to a cytokine storm, as measured by elevated levels of proinflammatory mediators, the rupture of critical structures in the lower respiratory tract, and hemorrhage [20,21]. We analyzed these parameters in animals infected and treated with 60 mg/kg of daclatasvir/day. The repurposed antiviral agent reduced cell death and lung necrosis, as indicated by the decreased levels of the intracellular marker LDH in bronchoalveolar lavage (BAL) fluid following treatment (Figure 3A), which also reduced cellular infiltration in bronchoalveolar (BAL) fluid (Figure 3B). Daclatasvir treatment prevented SARS-CoV-2-induced increases in the proinflammatory cytokines TNF-α, IL-6, and KC, both in the BAL fluid (Figure 3C–E) and in the lungs (Figure 3F–H).

In the lung histology of infected/untreated (Nil) mice, we observed diffuse alveolar collapse with multifocal septal rupture and intra-alveolar hemorrhage, along with moderate-to-severe interstitial edema and thickening of alveolar septa, resulting in a marked reductions in alveolar airspaces (Figure 4A). In Nil-treated mice compared to mock-infected mice, we detected the presence of numerous hyaline membranes lining alveolar spaces (yellow arrows, Figure 4A), the accumulation of proteinaceous debris and occasional cellular debris within alveolar spaces (blue arrows, Figure 4A), and multifocal aggregates of inflammatory cells, predominantly neutrophils and macrophages, within alveolar spaces and the interstitium (Figure 4A). Besides the histological score (Figure 4B), we estimated the extent of lesions to be equivalent to approximately 50% of the lung parenchyma, which is affected by alveolar collapse. On the bright side, the lungs of DCV-treated mice displayed a significant reduction in the severity and extent of alveolar collapse and septal rupture compared to nil-treated animals (Figure 4A,B). In DCV-treated mice, we only observed mild-to-moderate interstitial edema with minimal thickening of alveolar septa, reduced presence of hyaline membranes and proteinaceous debris within alveolar spaces, and minimal-to-mild inflammatory cell infiltration within alveolar spaces and the interstitium, and we estimated that approximately only 10% of the lung parenchyma was affected by alveolar collapse (Figure 4A,B). The level of protection observed in the lungs of the infected mice treated with 60 mg/kg/day of daclatasvir was associated with lower levels of double-stranded RNA (dsRNA), a biomarker of SARS-CoV-2 RNA synthesis (Figure 4C, amber-colored cells and/or red arrows for dsRNA) in the lungs.

### 3.3. Plasma and Lung Exposure to Daclatasvir at 60 mg/kg in Mice

Noninfected Swiss Webster outbred mice were treated with daclatasvir at 60 mg/kg to better interpret the in vivo efficacy results in light of daclatasvir’s pharmacokinetics and lung assessment. Thus, the compound was quantified both in the plasma and lungs at different times after its administration. To the best of our knowledge, this is the first assessment of daclatasvir levels in the respiratory tract. We observed that at 60 mg/kg, the plasma and lung exposure levels exceeded the in vitro pharmacological parameters to inhibit (by 50 and 90%, respectively, for the EC_50_ and EC_90_) SARS-CoV-2 replication in human pneumocytes, as previously described [12] (Figure 5). Importantly, daclatasvir was sustainably found (Figure 5 and Table 1). The maximum concentration (Cmax) and area under the curve (AUC) of daclatasvir in the plasma were approximately 3.3- and 5.9-fold greater, respectively, than those in the lungs (Table 1). Taken together, these results present a preclinical condition that supports the need for plasma and pulmonary exposure to daclatasvir to achieve anti-SARS-CoV-2 activity and mouse survival upon lethal infection.

## 4. Discussion

Drug repurposing was among the key strategies proposed by the World Health Organization (WHO) to fight the COVID-19 pandemic early in 2020. Although the solidarity trial led by the WHO did not show clinical benefit [22], more studies to better comprehend the mechanism of action and in vivo efficacy in light of the pharmacokinetic limitations of the repurposed drug are needed for translational science [23,24]. Our previous experience in repurposing atazanavir, an HIV protease inhibitor, against SARS-CoV-2 [25,26] and sofosbuvir, an HCV RNA polymerase inhibitor, against Zika [25,26], yellow fever [27], and chikungunya [28] shows also that translating preclinical data on repurposing into clinical trials may have strong limitations in terms of choosing the timing and trial design (doses, target population and outcomes) [29,30]. We recognize that to fulfill the ambition of phase II/III clinical trials from drug repurposing data, one must carry out this research as a precision medicine study, which requires that the proposed new use of a known drug follows its range of plasma exposure and body distribution to anatomical compartments where the targeted microorganism replicates. Alternatively, the new use of a clinically approved drug may be accompanied by a novel proposal of a regimen and dose, which would require a phase I clinical trial to validate the use of the repurposed drug. In this study, we carefully evaluated the efficacy of daclatasvir in a lethal SARS-CoV-2 mouse model aligned with a pharmacokinetic approach to improve our knowledge of whether this drug represents a chemical structure for further improvements in antiviral design and/or whether the data could be translated into new clinical trials.

We reported that daclatasvir, an inhibitor of the HCV NS5a protein, was endowed with anti-SARS-CoV-2 activity in human respiratory cell lines by targeting viral RdRp and ExoN activities [11,12], but these activities occurred at concentrations beyond human plasma exposure [12]. Nevertheless, the preclinical development of daclatasvir for the treatment of HCV was designed to determine the long-term safety aspects of this drug [14]. For diseases such as COVID-19, with a shorter acute phase, treatments are not supposed to be as long as they are for HCV, as exemplified by nirmatrelvir, remdesivir and molnupiravir authorized regimens, which take approximately one week. Despite the diverse clinical spectrum, which is categorized as long-term COVID-19, its potential use as an antiviral should be acute. In the preclinical pharmacopendium of daclatasvir, short-term treatments and single-dose and 28-day multidose treatments lead to plasma exposure in mice, rats, dogs, and nonhuman primates above the daclatasvir anti-SARS-CoV-2 activity described by us and others [13,14,31]. Therefore, we decided to further explore whether daclatasvir could protect hACE-2-K18 mice from lethal challenge with SARS-CoV-2 via doses that are equivalent to the preclinical doses that are considered safe in nonhuman primates [13,14,31].

At a dosage of 60 mg/kg/day, daclatasvir protected the mice from lethal challenge with SARS-CoV-2, reducing viral replication in the lung and consequently decreasing tissue insult and proinflammatory cytokine markers and increasing animal survival. In line with our previous works, which described daclatasvir targeting SARS-CoV-2 RNA polymerase and exonuclease [11,12], in the present study, we show in an animal model that this drug reduces RNA synthesis and triggers an “error catastrophe”, a process observed in viral populations, especially RNA viruses, where an overload of mutations during replication causes a breakdown in genetic integrity, eventually leading to the elimination of the virus [32,33,34,35].

Previous clinical trials on the use of daclatasvir against COVID-19 have shown conflicting results regarding whether this drug, at a standard anti-HCV dose of 60 mg/day, has clinical benefits [29,36,37,38]. In fact, the plasma and lung exposures of mice treated with 60 mg/kg daclatasvir were above the in vitro pharmacological parameters for SARS-CoV-2, and the drug penetrated well into the lungs. Considering this investigation, a new regimen may be envisioned. Nonhuman primates tolerate 15 mg/kg of daclatasvir [14], which is equivalent to 60 mg/kg of daclatasvir in mice and 5 mg/kg of daclatasvir in humans, according to protocols of dose conversion [39]. Considering that a body weight of 60 kg is usually used for pharmacokinetic assessment, the maximum daclatasvir dose rationalized in this study for new clinical trials would be 300 mg. Translating this dose into clinical trials against COVID-19 could lead to more consistent results. The dose of 60 mg/kg in mice leads to Cmax and plasma exposures similar to those observed in early clinical studies on daclatasvir development, such as when it was tested in humans at 200 mg/day [40] or at 120 mg/day [41], in combination with the antiretroviral drug enfavirenz. Owing to limited human experience with these high doses of daclatasvir, it would be more balanced to state that our work supports new phase I clinical trials of this generic drug to determine its pharmacokinetics, safety and tolerability over a few days at higher doses.

## 5. Conclusions

We previously demonstrated that DCV, an inhibitor of the HCV protein NS5A, impairs SARS-CoV-2 replication by targeting viral RNA polymerase and exonuclease at concentrations beyond the approved regimen against HCV. Because any potential use of DCV against SARS-CoV-2 would be shorter than that for HCV, we tested higher doses of these drugs, which are considerable tolerable for short-term toxicological studies. We found that K18-hACE-2 mice that were lethally infected with SARS-CoV-2 and treated with 60 mg/kg/day of DCV survived exhibited reduced virus replication and virus-induced lung insult in the lungs. Taken together, our data provide preclinical evidence that can support phase I clinical trials to confirm the safety, tolerability, and pharmacokinetics of new doses of daclatasvir for a short duration in humans to further advance this compound against COVID-19.

## Figures and Tables

**Figure 1 viruses-16-01856-f001:**
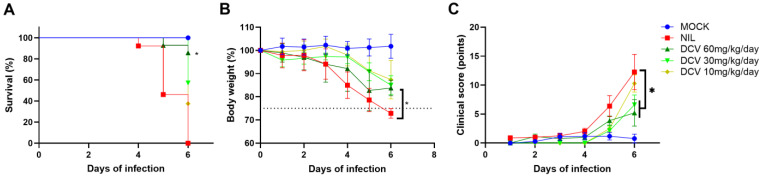
The effect of daclatasvir (DCV) on hACE2 mice infected with SARS-CoV-2. Transgenic hACE2 mice (10–12 weeks old) were intranasally infected with 10^5^ PFU of SARS-CoV-2 and treated via gavage with the indicated doses of DCV approximately 12 h after infection. The animals were observed daily for survival (**A**), changes in body weight (**B**), and clinical score (**C**). The clinical score included assessments of weight loss, reduced activity and exploration, eye closure or tearing, piloerection, posture, and respiration. Analyses were performed with at least 10 animals per experimental group; * *p* < 0.05,compared with SARS-CoV-2-infected/untreated (nil) animals. The dotted line in (**B**) represents a 25% decrease in body weight, which was considered the experimental endpoint at which to euthanize the mice to avoid suffering.

**Figure 2 viruses-16-01856-f002:**
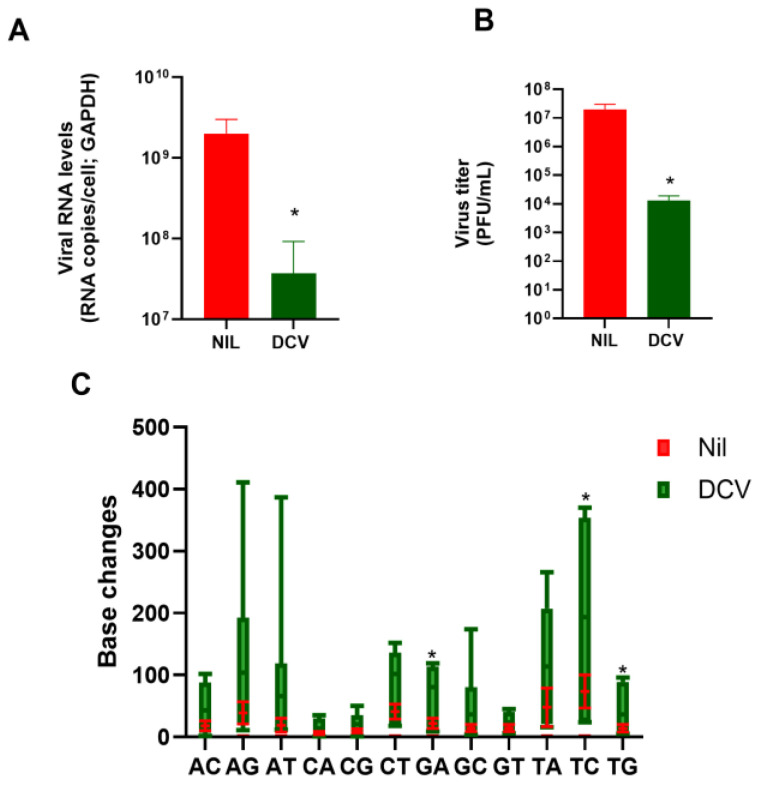
Daclatasvir reduced SARS-CoV-2 replication in the lungs of infected hACE2 mice. Transgenic hACE2 mice (10–12 weeks old) were infected with 10^5^ PFU of SARS-CoV-2 and treated with 60 mg/kg/day of daclatasvir (DCV) 12 h after infection. On the sixth day after infection, the animals were euthanized, and the lungs were collected. Viral RNA (**A**) and viral titers (**B**) were determined by quantitative RT–PCR and plaque assay (PFU/mL), respectively. Viral RNA was also subjected to unbiased sequencing via an MGI-g400 apparatus (**C**). All analyses were conducted with eight animals per experimental group; * *p* < 0.05 compared with SARS-CoV-2-infected/untreated (nil) animals.

**Figure 3 viruses-16-01856-f003:**
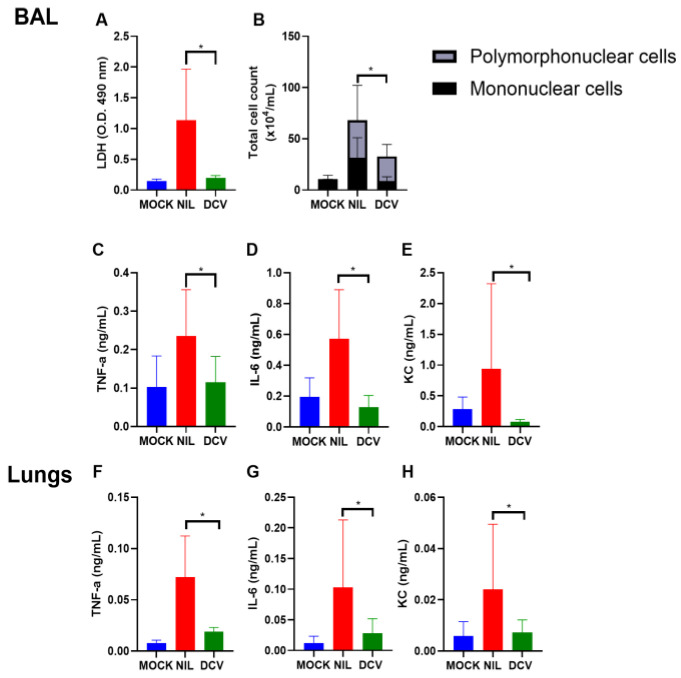
Daclatasvir reduces inflammation in hACE2 mice infected with SARS-CoV-2. Transgenic hACE2 mice (10–12 weeks old) were infected with 10^5^ PFU of SARS-CoV-2 and treated with 60 mg/kg/d of daclatasvir (DCV) 12 h after infection. On the sixth day postinfection, BAL fluid and lungs from euthanized mice were collected to measure LDH levels (**A**), polymorphonuclear and mononuclear cells were counted (**B**), and the levels of TNF-α (**C**,**F**), IL-6 (**D**,**G**), and KC (**E**,**H**) were measured. * Indicates *p* < 0.05.

**Figure 4 viruses-16-01856-f004:**
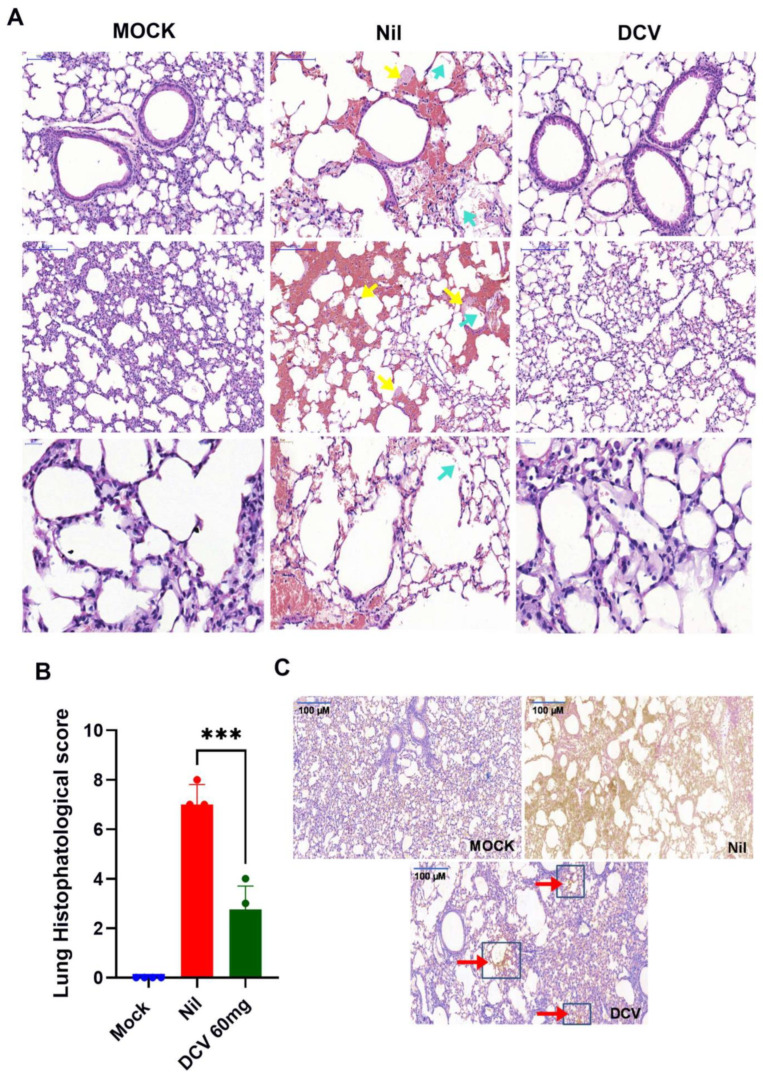
Representative histological assessment via hematoxylin and eosin (H&E) staining (**A**) (the presence of proteinaceous debris in the alveolar space = cyan arrows), (hyaline membrane formation = yellow arrows) of mouse lungs from the sixth day after infection. The histological scores of the lungs were determined after the mock, nil, and daclatasvir treatments (**B**). The immunohistochemistry results for dsRNA ((**C**), amber-colored cells indicated by red arrows/blue boxes for dsRNA) from three independent experiments are presented. Scale bar in (**A**) = 1000 µm. Scale bar in (**C**) = 100 µm. All analyses were performed with five animals per experimental group; *** *p* < 0.001 compared with SARS-CoV-2-infected and untreated animals (nil).

**Figure 5 viruses-16-01856-f005:**
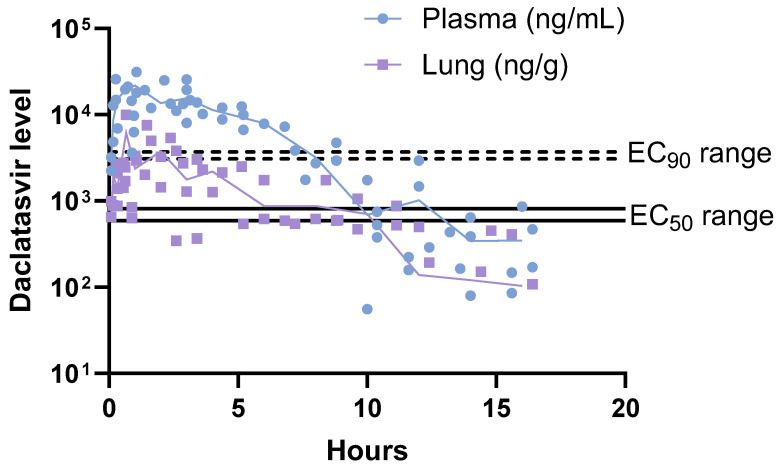
Pharmacokinetics of a single oral treatment with daclatasvir at 60 mg/kg orally. Five-week-old Swiss Webster mice were orally administered daclatasvir. At specified time points postadministration, the levels of daclatasvir were measured in both plasma and lung tissue. The lung samples were homogenized via an Ultra-Turrax Disperser (T-10 basic, IKA) for 30 s prior to analysis.

**Table 1 viruses-16-01856-t001:** The pharmacokinetic parameters based on the means of the regression curves.

Parameters	Plasma	Lung
λ	0.308 h*^−1^	0.0472 h*^−1^
T_1/2_	2.25 h	14.68 h
T_max_	1.00 h	0.67 h
C_max_	22,236.60 ng/mL	6186.26 ng/g
T_last_	16.00 h	16.00 h
C_t_	353.84 ng/mL	103.46 ng/g
AUC_last_	101,450 h.ng/mL	16,932 h.ng/g
AUC_∞_	102,596.81 h.ng/mL	19,123 h.ng/g

## Data Availability

Data supporting the findings of this study are available from the corresponding author, T.M.L.S., on reasonable request. The sequences generated in this study are available via the Bioproject accession code PRJNA1161613.

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
