# Peer review of "Newly Proposed Dose of Daclatasvir to Prevent Lethal SARS-CoV-2 Infection in Human Transgenic ACE-2 Mice"

_viruses, 2024, doi:10.3390/v16121856_

Round 1
Reviewer 1 Report
Comments and Suggestions for Authors
The manuscipt by Mattos and co-authors reports on the appliation of daclatasvir, an inhibitor of the RdRp and ExoN of SARS-CoV-2 in a well established mouse model of SARS-CoV-2 infetion, the K18-hACE2 mouse. The study provides interesting data, but suffers severely from the lack of an appropriate histological examination and a meaningful in situ assessment of the extent of pulmonary virus infection
Materials and Methods:
Line 107: More information on the virus isolate should be provided, at least an appropriate reference. It is not clear why reference 15 is cited in the following sentence, as there is no evidence that the referenced paper used the same virus isolate.
Line 117: Information is missing that the animals were euthanised and how this was done.
Line 214: The abbreviations “KC” and “PF4” need to be introduced.
Line 223: The authors state that “morphological alterations in the (lung) tissue were obsereved and documented”. No information is provided with regards to a potential “Histopathological Score”. However, this is then presented without explanation in Figure 4B, hence this entirely obscure and meaningless/arbitrary.
Line 226: “appropriate preimmune serum” should be specified.
Results:
Line 287/288: It is not appropriate to refer to histological changes in the lung of human patients with COVID-19 for the assessment of SARS-CoV-2 infection in the K18-hACE2 mouse model. A description of the typical changes in this mouse model could be found in reference 15 (and innumerous other publications). In any case, haemorrhage is not a relevant feature in the lungs of mice, different from what the authors seem to believe.
The figure legends (Fig. 3 and 4) should contain information on the day post infection on which the animals were examined, as this determines the changes and extent of viral antigen expression that can be expected in infected untreated mice.
The results of the histological examination seem to be reported in lines 283 and 284. Here, the authors mention that dacatasvir treatment “preserved lung integrity and significantly prevented the hemorrhage observed in infected mice”. This statement is inappropriate as a description of the histopathological changes induced by SARS-CoV-2 in the lungs of K18-hACE2 mice. It is likely that this reflects the lack of involvement of a pathologist in general, and in particular of a pathologist with knowledge on the histopathological features associated with SARS-CoV-2 infection of the lung in the mouse.
The authors also mention a “reduction in lung damage” and “lower levels of double stranded RNA (dsRNA), a biomarker of SARS-CoV-2 synthesis” (lines 286-287). This type of superficial reporting of histological features and their interpretation is not acceptable.
Figure 3: The legend claims that “daclatasvir reduced inflammation, hemorrhage and preserved lung integrity in mice infected with SARS-CoV-2”. However, this is not shown in this figure which only shows leukocytes in the BAL and cytokines etc. in BAL and lungs.
Figure 4: This is incorrectly labelled (I, J, K instead of A, B, C). No information is given as to the day post infection at which the mice were examined, and no information is given on the histological changes and .
Also, the staining for dsRNA cannot be assessed. There is an almost diffuse beige staining reaction which cannot reflect the presence of virus, as this is restricted to alveolar epithelial cells. The staining of these cells cannot be assessed as the images are of too low resolution and magnification. For the DCV treated animal, something is apparently highlighted by squares in the image. However, this cannot be assessed as, again, the resolution is poor and the magnification too low.
Furthermore, both negative and positive controls seem to lack for the dsRNA staining (neither is mentioned in M&M). These should be included, in particular since the authors have used a mouse monoclonal antibody against dsRNA in immunohistology. They claim that this is as “a biomarker of SARS-CoV-2 synthesis”. However, it appears that the mouse-on-mouse staining was not optimal. The use of the chosen antibody is therefore questionable, and application of one of the well-published (rabbit-) anti SARS-CoV-2 NP antibodies would probably be the better option, since this would also be evidence of viral replication.
Author Response
POINT-BY-POINT RESPONSE
Reviewer 1
The manuscipt by Mattos and co-authors reports on the appliation of daclatasvir, an inhibitor of the RdRp and ExoN of SARS-CoV-2 in a well-established mouse model of SARS-CoV-2 infetion, the K18-hACE2 mouse. The study provides interesting data, but suffers severely from the lack of an appropriate histological examination and a meaningful in situ assessment of the extent of pulmonary virus infection
Materials and Methods:
Line 107: More information on the virus isolate should be provided, at least an appropriate reference. It is not clear why reference 15 is cited in the following sentence, as there is no evidence that the referenced paper used the same virus isolate.
The reviewer is correct; we removed this citation. The SARS-CoV-2 gamma variant, also referred to as the P1 lineage, was obtained from a nasopharyngeal specimen of a patient from Manaus city, Brazil, in December 2020 and is accessible via EPI_ISL_1060902, hCoV-19/Brazil/AM-L70-71-CD1739/2020. The isolate was grown in Vero E6 cells, which were used for SARS-CoV-2 propagation and titration.
Line 117: Information is missing that the animals were euthanised and how this was done.
Indeed, we missed this information in the previous version. Thank you for pointing that out. We added this information to the Materials and Methods subsection 2.4. Please see the change in red, which now reads: “Euthanasia was performed to alleviate animal suffering in cases of weight loss > 25%. For the collection of biological material, on the 6th day postinfection, euthanasia was performed subcutaneously (dorsal-anterior region of the animal) with a dose of 150 mg/kg ketamine and 10 mg/kg xylazine in a volume of 70 µL (using an insulin syringe with an 8 mm needle). Once the animal is fully anesthetized (pain sensitivity test by lightly pressing the paw), blood will be collected via cardiac puncture (using a 3 mL syringe and a 22G needle). After the completion of blood collection by cardiac puncture, the animals received a lethal dose of anesthetics (300 mg/kg ketamine and 30 mg/kg xylazine), and cardiac‒respiratory arrest was confirmed via a stethoscope.”
Line 214: The abbreviations “KC” and “PF4” need to be introduced.
We added this information to the Materials and Methods subsection 2.9. Please see the changes in red, which now reads: “KC (Keratinocyte-derived Cytokine) and PF4 (Platelet factor 4).”
Line 223: The authors state that “morphological alterations in the (lung) tissue were obsereved and documented”. No information is provided with regards to a potential “Histopathological Score”. However, this is then presented without explanation in Figure 4B, hence this entirely obscure and meaningless/arbitrary.
We made this clearer now because we included more details in subsection 2.10 of the Materials and Methods section. The updated content is marked in red for your review: "The morphological alterations observed in the lung tissue were assessed using an inflammatory scoring system: (i) airway inflammation (maximum of 4 points), (ii) vascular inflammation (maximum of 4 points), (iii) parenchymal inflammation (maximum of 5 points), and overall neutrophil infiltration (maximum of 5 points) (https://pubmed.ncbi.nlm.nih.gov/34495692/).
Line 226: “appropriate preimmune serum” should be specified.
Thank you for bringing this to our attention. We have now included this information in subsection 2.10 of the Materials and Methods, marked in red: “Normal guinea pig serum (Sigma‒Aldrich, cat # 566400)”.
Results:
Line 287/288: It is not appropriate to refer to histological changes in the lung of human patients with COVID-19 for the assessment of SARS-CoV-2 infection in the K18-hACE2 mouse model. A description of the typical changes in this mouse model could be found in reference 15 (and innumerous other publications). In any case, haemorrhage is not a relevant feature in the lungs of mice, different from what the authors seem to believe.
As described by Oladunni et al. (https://pmc.ncbi.nlm.nih.gov/articles/PMC7705712/) and Dong et al. (https://pmc.ncbi.nlm.nih.gov/articles/PMC8754221/), hemorrhage in the lungs of transgenic k18-hACE2 mice infected with SARS-CoV-2 is a well-established characteristic of this experimental model, which further validates the successful reproduction of this lethal model in our study.
The figure legends (Fig. 3 and 4) should contain information on the day post infection on which the animals were examined, as this determines the changes and extent of viral antigen expression that can be expected in infected untreated mice.
These legends were adjusted; please see the changes in red below and in the manuscript.
Figure 3. Daclatasvir Reduces Inflammation in hACE2 Mice Infected with SARS-CoV-2 Transgenic hACE2 mice (10–12 weeks old) were infected with 105 PFU of SARS-CoV-2 and treated with 60 mg/kg/d daclatasvir (DCV) 12 h after infection. On the 6th day postinfection, BAL fluid and lungs from euthanized mice were collected to measure LDH levels (A), polymorphonuclear and mononuclear cells were counted (B), and the levels of TNF-α (C and F), IL-6 (D and G), and KC (E and H) were measured.
Figure 4. Representative histological assessment via hematoxylin and eosin (H&E) staining (I) of mouse lungs from the 6th day after infection. Histological scores of the lungs were determined in the mock, nil and daclatasvir groups (J). The immunohistochemistry results for dsRNA (K, amber-colored cells for dsRNA) from three independent experiments are presented. Scale bar in (I) = 1,000 µm. Scale bar in (K) = 100 µm. All analyses were performed with 5 animals per experimental group; * p<0.05 compared with SARS-CoV-2-infected and untreated animals (nil).
The results of the histological examination seem to be reported in lines 283 and 284. Here, the authors mention that daclatasvir treatment “preserved lung integrity and significantly prevented the hemorrhage observed in infected mice”. This statement is inappropriate as a description of the histopathological changes induced by SARS-CoV-2 in the lungs of K18-hACE2 mice. It is likely that this reflects the lack of involvement of a pathologist in general, and in particular of a pathologist with knowledge on the histopathological features associated with SARS-CoV-2 infection of the lung in the mouse. The authors also mention a “reduction in lung damage” and “lower levels of double stranded RNA (dsRNA), a biomarker of SARS-CoV-2 synthesis” (lines 286-287). This type of superficial reporting of histological features and their interpretation is not acceptable.
We improved the description of the histopathological findings by including the following in the manuscript: “Lung histology of infected/untreated (Nil) mice revealed collapsed alveoli with ruptured septa, increased wall thickness with decreased air space, notable hemorrhage, proteinaceous debris in the alveolar space (blue arrows), hyaline membrane formation (yellow arrows), and disordered parenchyma in the tissue. The lung parenchyma of the treated animals more closely resembled that of the mock-infected mice, and a significant improvement in the previously noted findings was observed. (Figure 4A and B). Decreased lung damage was observed in infected mice treated with 60 mg/kg/day daclatasvir, which was associated with lower levels of double-stranded RNA (dsRNA), a biomarker of SARS-CoV-2 RNA synthesis that is produced during various viral infections, and its immunohistochemical detection has been proposed as a potential marker for identifying viral replication. (Figure 4C, amber-colored cells and/or red arrows for dsRNA), in the lungs.”
Figure 3: The legend claims that “daclatasvir reduced inflammation, hemorrhage and preserved lung integrity in mice infected with SARS-CoV-2”. However, this is not shown in this figure which only shows leukocytes in the BAL and cytokines etc. in BAL and lungs.
Thank you for pointing that out. The change has been made to "Daclatasvir Reduces Inflammation in hACE2 Mice Infected with SARS-CoV-2.”
Figure 4: This is incorrectly labelled (I, J, K instead of A, B, C). No information is given as to the day post infection at which the mice were examined, and no information is given on the histological changes. Also, the staining for dsRNA cannot be assessed. There is an almost diffuse beige staining reaction which cannot reflect the presence of virus, as this is restricted to alveolar epithelial cells. The staining of these cells cannot be assessed as the images are of too low resolution and magnification. For the DCV treated animal, something is apparently highlighted by squares in the image. However, this cannot be assessed as, again, the resolution is poor and the magnification too low. Furthermore, both negative and positive controls seem to lack for the dsRNA staining (neither is mentioned in M&M). These should be included, in particular since the authors have used a mouse monoclonal antibody against dsRNA in immunohistology. They claim that this is as “a biomarker of SARS-CoV-2 synthesis”. However, it appears that the mouse-on-mouse staining was not optimal. The use of the chosen antibody is therefore questionable, and application of one of the well-published (rabbit-) anti SARS-CoV-2 NP antibodies would probably be the better option, since this would also be evidence of viral replication.
During viral replication, viruses generate reverse-oriented copies of their genome or mRNA to exploit the host cell machinery and produce progeny. This process results in the formation of double-stranded RNA (dsRNA) intermediates, which are formed by the complementary binding of negative- and positive-sense RNA strands. For SARS-CoV-2, dsRNA acts as evidence of active viral replication, particularly in single-stranded RNA (ssRNA) viruses (https://insight.jci.org/articles/view/139042; https://www.nature.com/articles/s41598-020-78949-0). The presence of dsRNA intermediates is an indicator of viral replication (https://pubmed.ncbi.nlm.nih.gov/8638399/, https://journals.asm.org/doi/10.1128/jvi.80.10.5059-5064.2006). Consequently, anti-dsRNA antibodies are valuable tools for assessing panviral replication in situ. Similar to other studies, dsRNA labeling with J2 clones was restricted to the perinuclear cytoplasm, which corresponds to the replication site of coronaviruses (https://www.nature.com/articles/s41598-020-78949-0). Uninfected mice were used as controls, and no J2 strain was detected. Infected and untreated mice constitute another control with strong labeling.
Reviewer 2
This manuscript is a well-written, compact, but rationally structured study of the possibility of repurposing a known drug daclatasvir, a clinically approved inhibitor of hepatitis C virus (HCV). The authors propose using higher than approved doses for a shorter period of time than is used in HCV treatment. The data support the idea that such a regimen could be investigated in more detail (clinical studies). The manuscript deserves publication in Viruses MDPI
Below are the reviewer's suggestions:
Line 127: «Lungs were assessed for viral load via quantitative RT‒PCR and PFU/mL, as previously described [19].» However, this publication is a review article and there are no methods for determination of viral titer (PFU/ml). Please, describe the plaque assay in details.
Thank you for bringing that to our attention. The following adjustment was made: “Lungs were assessed for viral load via quantitative RT‒PCR and virus titration, Vero cells (2.0 × 104 cells/well) in 96-well plates (Nalge Nunc Int, Rochester, NY, USA) were infected with serial log-based dilutions of supernatants from the lungs for 1 hour at 37 °C with 5% CO2. Following incubation, medium containing 1.8% CMC and 5% FBS was added, and the cells were incubated at 37°C with 5% CO2 for 72 hours. The cells were then fixed with 10% formaldehyde in PBS and stained with a 0.04% crystal violet solution in 70% methanol. Virus titers were determined by counting the plaque-forming units (PFU/mL) (https://pubmed.ncbi.nlm.nih.gov/35056078/). In addition, histological analysis, unbiased sequencing and metatranscriptomic approaches were performed.”
Line 160: «tert-methyl-butyl-ether (TBME)» should be tert-butyl methyl-ether (TBME).
Thank you for the valuable suggestion; we have made the change as requested: “tert-butyl methyl-ether (TBME)”.
Line 247: We thus infected hACE2 MICE with SARS-CoV-2 intranasally and treated the animals daily via oral gavage with daclatasvir at 10, 30 and 60 mg/kg.
Thank you for the valuable suggestion; we have made the change as requested: “We thus infected hACE2 MICE with SARS-CoV-2 intranasally and treated the animals daily via oral gavage with daclatasvir at 10, 30 and 60 mg/kg”.
Although the conclusion is present implicitly in the discussion section, the reviewer advises to add the conclusion as a separate section.
Thank you for this suggestion.

Reviewer 2 Report
Comments and Suggestions for Authors
This manuscript is a well-written, compact, but rationally structured study of the possibility of repurposing a known drug daclatasvir, a clinically approved inhibitor of hepatitis C virus (HCV). The authors propose using higher than approved doses for a shorter period of time than is used in HCV treatment. The data support the idea that such a regimen could be investigated in more detail (clinical studies). The manuscript deserves publication in Viruses MDPI
Below are the reviewer's suggestions:
Line 127: «Lungs were assessed for viral load via quantitative RT‒PCR and PFU/mL, as previously described [19].»
However, this publication is a review article and there are no methods for determination of viral titer (PFU/ml). Please, describe the plaque assay in details.
Line 160: «tert-methyl-butyl-ether (TBME)» should be tert-butyl methyl-ether (TBME)
Line 247: We thus infected hACE2 MICE with SARS-CoV-2 intranasally and treated the animals daily via oral gavage with daclatasvir at 10, 30 and 60 mg/kg.
Although the conclusion is present implicitly in the discussion section, the reviewer advises to add the conclusion as a separate section.
Author Response

(The authors gave the same response as above.)

Round 2
Reviewer 1 Report
Comments and Suggestions for Authors
The manuscript has improved, mainly because it now includes esssential information that should have been provided in the first instance and not after prompting by reviewers.
Unfortunately, the manuscript has not really improved with regards to the morphological aspects of the study. The papers listed as references regarding alveolar haemorrage as an essential feature to assess the extent of lung damage/changes in the K18 mouse model mention alveolar haemorrhage as one feature. It is, however, not of the relevance that the authors claim. There had been easy ways to better determine the extent of alveolar damage.
The histological descriptions provided in the revised manuscript read very much like bits of the descriptions in the references, a bit like a copy-paste exercise, apart from the term "disordered parenchyma" which is definitely not a term used for a histopathological description - and meaningless...
Most parameters for scoring (airway inflammation, vascular inflammation, parenchymal inflammation) are too vague and too poorly defined to be meaningful. It is preferable not to include any scoring than including such a makeshift scoring that will not have provided reproducable results.
Thanks for making the effort to explain the occurrence of dsRNA in SARS-CoV-2 replication; this was no news to the reviewer. The issue is not that the reviewer was not aware of the potential usefulness of such a staining approach, the issue is the staining result which, unfortunately, is not more convincing than before in the images that have been provided (low resolution, low magnification, diffuse beige staining...).
The reviewer does not question the effect of the tested compound, several parameters are well examined, quantified and presented. It is the morphological work-up, i.e. the description and scoring of the pulmonary histological changes as well as the immunohistological work-up of poor quality that does reduce the quality and value of the manuscript.
Author Response
Methods:
We made additional changes to lines 240-242.
We explicitly mention the reference: Instead of just putting a number at the end, directly state "PMID: 34495692" within the sentence. This makes it crystal clear where the scoring system comes from.
"Previously published": This adds context and clarifies that you're not introducing a new scoring system.
Results:
We improved the descriptions, by enhancing the descriptions and incorporating more precise terminology and quantifiable measures, such as:
For NIL:
- Diffuse alveolar collapse with multifocal septal rupture and intra-alveolar hemorrhage.
- Moderate to severe interstitial edema and thickening of alveolar septa, resulting in a marked reduction of alveolar airspaces.
- Presence of numerous hyaline membranes lining alveolar spaces (yellow arrows).
- Accumulation of proteinaceous debris and occasional cellular debris within alveolar spaces (blue arrows).
- Multifocal aggregates of inflammatory cells, predominantly neutrophils and macrophages, within alveolar spaces and interstitium.
- Quantify the extent of the lesions (e.g., "approximately 50% of the lung parenchyma is affected by alveolar collapse").
For DCV:
- Significant reduction in the severity and extent of alveolar collapse and septal rupture compared to NIL.
- Mild to moderate interstitial edema with minimal thickening of alveolar septa.
- Reduced presence of hyaline membranes and proteinaceous debris within alveolar spaces.
- Minimal to mild inflammatory cell infiltration within alveolar spaces and interstitium.
- Quantify the improvement (e.g., "approximately 10% of the lung parenchyma is affected by alveolar collapse").
Final comments:
We expect that, by using more specific and quantitative language, the improved descriptions provide a clearer and more comprehensive understanding of the pathological changes observed in the lungs of the mice.
We kindly ask for this Reviewer to be respectful and do his/her best to respond from scientist to scientist and improve the quality of work without conflicts of interest. Some of your words are positive, such as: "The reviewer does not question the effect of the tested compound, several parameters are well examined, quantified and present" and "The manuscript has improved, mainly because it now includes esssential information…". However, terms that suggest plagiarism “copy-paste exercise”, and that we did not carry out our research with scientific rigor “...will not have provided reproducable results” should be clearly proven. The rigor and quality of our research with daclatasvir is the same used on other publications, such as Preclinical development of kinetin as a safe error-prone SARS-CoV-2 antiviral able to attenuate virus-induced inflammation | Nature Communications.
Reviewer 2 Report
Comments and Suggestions for Authors
The authors took into account the suggestions of the reviewer
Please correct as follows: tert-butyl methyl ether
Author Response
We removed the hyphen between methyl and ether in the line 176